# Evaluation of Tooth Movement Accuracy with Aligners: A Prospective Study

**DOI:** 10.3390/ma15072646

**Published:** 2022-04-04

**Authors:** Vincenzo D’Antò, Rosaria Bucci, Vincenzo De Simone, Luis Huanca Ghislanzoni, Ambrosina Michelotti, Roberto Rongo

**Affiliations:** 1Department of Neuroscience, Reproductive Sciences and Oral Sciences, University of Naples “Federico II”, 80138 Naples, Italy; rosaria.bucci@unina.it (R.B.); vincenzo.desimone.aza@gmail.com (V.D.S.); michelot@unina.it (A.M.); roberto.rongo@unina.it (R.R.); 2Department of Orthodontics, Dental School, University of Geneva, 1205 Geneva, Switzerland; luis.huancaghislanzoni@unige.ch

**Keywords:** aligners, clear aligner treatment, digital dentistry, treatment accuracy, tooth movement

## Abstract

Background. Clear aligners treatment (CAT) is a common solution in orthodontics to treat both simple and complex malocclusions. This study aimed to evaluate the predictability of CAT, comparing the virtually planned and the achieved tooth movement at the end of stage 15, which is often the time of first refinement. Methods. Seventeen patients (mean age: 28.3 years) were enrolled in the study. Torque, tip and rotation were analyzed in 238 maxillary teeth on digital models at Pre-treatment (T0), at the end of stage 15 (T15) and at virtually planned stage 15 (T15i). Prescription, Achieved movement and performance values were calculated to compare the virtually planned and the clinical tooth position. Data were analyzed by means of Student’s *t* test with a level of significance set at *p* < 0.05. Results. The largest iper-performance was the torque correction of the second molars (+2.3° ± 3.1°), the greatest under-performance was the tip correction of the first molars (−2.3° ± 3.3°), while rotation corrections of all the teeth showed more accurate performance. No significant differences were found between mean Prescription and mean Achieved movement for all the assessed movements (*p* < 0.05). Conclusions. An accurate evaluation of CAT after the 15th aligner is fundamental in order to individuate the movements that are not matching the digital set-up.

## 1. Introduction

The increasing concern of orthodontic patients regarding the aesthetic impact of their treatments has driven the introduction of new appliances or strategies to meet this rising aesthetic demand [1]. Clear aligners are thermoplastic orthodontic appliances that can generate a customized tooth movement thanks to the introduction of the CAD/CAM technology in orthodontics [2]. In 1946, Kesling had the first intuition to use thermoplastic devices for orthodontic purpose and, several decades later, Align Technology© (Santa Clara, CA, USA) was the first company that started to produce clear aligners [3]. After their introduction as an effective tool to manage from simple to mild malocclusions, clear aligners therapy (CAT) has become, in contemporary practice, a valid alternative to fixed labial or lingual orthodontics to treat even complex cases such as space closure [4,5,6].

The treatment with aligners is based on a step approach: the orthodontic tooth movement is virtually planned on a three-dimensional model of the patient’s malocclusion, and a sequence of thermoplastic devices is produced to obtain dental movements.

The worldwide diffusion of CAT is increasing mainly due to the attractiveness of this system for both patients and clinicians [7], there is not a solid evidence-based consensus in the literature on the predictability of different tooth movements with aligners [8,9,10] or on their effect on root resorption [11] or oral hygiene [12]. Comparing the virtual set-up with the real dental changes, several authors concluded that mesio-distal tipping is the most predictable movement, whereas extrusion and teeth rotations are the hardest movements to achieve [13,14]. The accuracy of movements with clear aligners is reported on average from 55% to 72% [15], with lower accuracy achieved by canine rotation (less than 36%) [16].

However, the effectiveness of tooth movement is strictly related not only to the virtual setup but also to the mechanical properties of the thermoplastic materials of aligners and attachment design [9,17,18]. Indeed, hardness, elasticity, resilience and resistance to storage in artificial saliva are related to tooth movements [17]. Furthermore, the use of aligners in the oral cavity exposes the aligner to other factors such as temperature, humidity, salivary enzymes and elastic deformation that could influence the physical and chemical properties of clear aligners in the mouth of patients [19,20]. Aligners are affected by intraoral aging that does not change the chemical composition but deteriorate the mechanical properties [21]. Therefore, this behavior can strongly influence the clinical outcome of CAT. Indeed, CAT has a fast and important drop of the force delivered by the single aligner due to its stress-relaxation property and intraoral degradation [22]. Considering this poor predictability, many clinicians reported that more than half of the aligner cases required refinements, corrections or fixed appliances [22]. A common time for the first refinement is around 8–12 months of treatment [23], for this reasons, in this study, it was decided to assess the tooth movements after stage 15.

Hence, the aim of this study was to evaluate the predictability of clear aligners not at the end of treatment but comparing the virtually planned and the achieved tooth movement at the end of the stage 15, exclusively in the maxillary arch. The null hypothesis is that no difference exists between the clinically registered movement and the virtual set-up.

## 2. Materials and Methods

The study was designed as a prospective single-center study. The study group was composed by patients referred to the School of Orthodontics of the Department of Neurosciences, Reproductive Sciences and Oral Sciences at the University of Naples Federico II (Italy) between September 2017 and November 2018. The first consultation was performed by one expert operator (VD) to evaluate whether they were suitable for CAT orthodontic treatment. Each patient signed specific informed consent. The study was conducted in accordance with the Declaration of Helsinki and approved by the Institutional Review Board (or Ethics Committee) of the University of Naples Federico II Prot. 353/19. The inclusion criteria to be enrolled in the study were the following:Adult patients (>18 years) who had no previous orthodontic treatment;Absence of local and systemic conditions or ongoing pharmacological treatment that can affect the tooth movement process;Non-extractive orthodontic treatment;Crowding up to 7 mm per arch;Absence of tooth shape anomalies;Absence of supernumerary teeth;Absence of tooth rotation more than 35°;Spaces up to 7 mm per arch;Good oral hygiene with absence of an active periodontal disease.

### 2.1. Orthodontic Treatment Protocol

All patients were treated by means of clear aligners provided by Airnivol^®^ (Airnivol^®^ s.r.l., Cascina, Italy) at the School of Orthodontics at the University of Napoli “Federico II”. The orthodontic treatment setup and the staging of the aligners were planned according to the limits of the following tooth movements at each stage:2° for rotation.2.5° for mesio-distal and buccal-lingual tip.0.25 mm for linear displacements.

The orthodontic treatment was performed using clear aligners and attachments. Bonded composite attachments on selected teeth were used, if needed, to improve tooth movements [24]. Moreover, Class II or Class III elastics as well as Inter Proximal Reduction (IPR) were planned, if required, for the resolution of the malocclusion. Patients were recommended to wear the aligners for a minimum of 22 h a day except during meals and during oral hygiene procedures and to replace them every 10 days.

### 2.2. Study Protocol: Measurements of Digital Model

Pre-treatment (T0) and at the end of the stage 15 (T15) digital models of the maxillary arch of each patient enrolled in the study were acquired using an intraoral scanner (CS 3600, Carestream Dental LLC, Atlanta, GA, USA) and stored in stl format. In addition, ideal digital models at stage 15 (T15i) were produced in stl format from the virtual set-up. T0, T15 and T15i digital models were than analyzed and compared by a single operator (VDS) using VAM software (Vectra, Canfield Scientific, Fairfield, NJ, USA).

Tooth position in each digital model from the central incisors to the second molars was analyzed by placing 72 reference points per arch (Figure 1), according to the method of L. Huanca Ghislanzoni et al. [25].

For each tooth, five points were taken: the mesial and the distal point of the occlusal surface, the gingival and the occlusal limit of the vestibular facial axis clinical crown (FACC) and the gingival limits of the palatal FACC. The point coordinates (XYZ) were exported as a txt file and then imported into a spreadsheet provided by the software. To determine the mesio-distal (M-D) angulation (tip), the buccal-lingual (B-L) angulation (torque) and the rotation of each tooth an occlusal reference plane was constructed considering the following points:The tip of the mesio-buccal cusp of tooth 16.The tip of the mesio-buccal cusp of tooth 26.The centroid of all occlusal points of the FACC of teeth 15, 14, 12, 11, 21, 22, 24 and 25; canines will be excluded from this calculation as their occlusal FACC point is generally outside the occlusal plane identified by the other teeth.

### 2.3. Study Variables: Analysis of Prescription, Achieved Tooth Movement and Performance

The value of torque, tip and rotation was determined for all teeth in the maxillary arches of the patients enrolled in the study for a total of 294 teeth. For all teeth and for all the three typologies of movement, study variables included “Prescription”, “Achieved movement” and “Performance”, as described below:***Prescription*** was calculated as the difference between ideal post-treatment (T15i) and pre-treatment (T0) measurements, to identify the amount in degrees of the planned movement:***Absolute Prescription*** = |ideal posttreatment−pretreatment|;***Achieved movement*** was calculated as the difference between real post-treatment (T15) and pre-treatment (T0) measurements:***Absolute Achieved movement*** = |real posttreatment—pretreatment|***Performance*** was calculated as the difference between Achieved movement and Prescription:***Absolute Performance*** = |Achieved movement − Prescription|

The closer the value to 0 of the performance, the more precise the dental movement produced by the aligners: a positive value indicates an iper-correction while a negative value indicates an under-correction. However, for all the included variables, the absolute values were considered to underline the accuracy of the movement without the influence of the direction of the movement (clockwise vs. anticlockwise rotation and lingual vs. buccal for the torque or mesial vs. distal for the tip).

The ***Frequencies of Performance Error (FOPE)*** was calculated as the frequencies of absolute values of Performance lower than or equal to 1°; between 1.1° and 2°, between 2.1° and 4°; higher than or equal to 4.1°, the FOPE was calculated for each movement for each tooth. Finally, for all the assessed movements, the frequencies of under-performance (Performance < −1°), right-performance (−1° ≤ Performance ≤ 1°) and iper-performance (Performance > 1°) were analyzed.

### 2.4. Statistical Analysis

Sample size calculation was performed considering the difference between Achieved movement and Prescription as primary outcome. Considering a *t* test for paired data and an average effect size of 0.5, an alpha error of 0.5 and a beta error of 0.2, 17 subjects for each group were needed [14]. Descriptive statistics and inferential analysis were performed. After 4 weeks, points selection on digital models was performed and repeated by the same operator on five randomly selected digital models, the Dahlberg’s Formula [26] and a student’s *t* test for paired data was used to identify the method of error and systematic error (*p* < 0.05). The Shapiro–Wilk test was performed to assess the distribution of the data. A Student’s *t* test for paired data was used to evaluate if any statistically significant difference was present between Absolute Prescription and Absolute Achieved movement. A Chi-Square test was used to evaluate the distribution of under-performance, right-performance and iper-performance among the three movements, with *p* set as <0.05.

## 3. Results

The study included 17 patients (11 females and 6 males), with a mean age of 28.3 years.

The method of error and the systematic error were shown in Table 1, there was not systematic error among all the assessed variables (*p* < 0.05), and the method of error ranges between 0.26 and 3.82.

Table 2 shows the mean values and standard deviation for the Absolute Prescription and Absolute Achieved movement for the torque, tip and rotation of all the teeth considered in the study. The largest planned movement was the tip correction of the first molars (5.7° ± 11.1°). The smallest planned movement was the torque correction of the canine (2.4° ± 2.2°). No significant differences were found between Absolute Prescription and Absolute Achieved movement for all the movements.

Table 3 shows the mean performance calculated as absolute values for each type of tooth and each kind of tooth movement and the FOPE. The largest deviation from the right performance detected was the tip of the second molars (3.8° ± 2.7°), that was also the movement showing the highest FOPE with 15 out of 34 (44.1%) performance with an error of more than 4°. The lowest deviation was shown by the rotation of the central incisors (1.8° ± 1.4°) while the movements that showed the highest frequencies of error of less than 1° were the torque of central incisors and canine (12; 35.3%).

Table 4 shows the distribution among different tooth movements of under-performance, right performance and iper-performance, analyzed by means of chi-square test. Among the three movements, the tip showed the highest frequency of under-performance (112/238, *p* < 0.001) while the torque showed the highest frequency of iper-performance (110/238, *p* < 0.001).

## 4. Discussion

The aim of the present study was to evaluate the predictability of orthodontic movement of the maxillary teeth with the Airnivol^®^ aligner system in terms of torque, tip and rotation. Specifically, the discrepancy between the virtually planned and the clinically achieved position of each tooth was evaluated and a performance index was calculated to identify the accuracy of movements made only with aligners. The orthodontic treatment in the study group was performed in adult patients (>18 years) with high motivation to start an orthodontic treatment because most of the therapeutic success depends on the home compliance of the patient [27]. Dahlberg’s formula presents a good reproducibility of the method, this result is consistent with the outcome of the study performed by Huanca Ghislanzoni et al. [25].

Even if aligners are becoming a more frequently used orthodontic device in the contemporary practice of treating mild to complex cases, the evidence from the literature about the predictability of this systematic is weak [6,28]. In order to evaluate the predictability as well as the accuracy of the planned tooth movements compared with those clinically achieved, a major factor that has a huge impact on the results reliability is the methodological procedure that is used to analyze mismatching between digital models. In our study, the method proposed and validated by Huanca Ghislanzoni et al. was used since it has a big advantage of evaluating the changes in terms of degrees of torque, tip and rotation of each tooth, having no measurement or systematic errors greater that 2° [25]. In our study, teeth were grouped separately in the data analysis since the response of the aligner treatment can vary in the different types of tooth because of the different anatomy of crowns and roots [15]. The results, indeed, showed a variability especially for the performance index, pointing out that CAT efficacy in delivering force and moments also depends on tooth shape and anatomy.

In order to give more clinical relevance to our results, a threshold of 1.0° can be considered in evaluating the performance index [29]. Although, no significant differences were found on average between Absolute Prescription and Absolute Achieved movement for all the movements, the performance index and the FOPE showed relevant results. The tip of the second molars (3.8° ± 2.7°) showed the largest deviation from the right prescription and in 15 out of 34 measurements (44.1%) the Achieved movements differed from the Prescription of more than 4°. Furthermore, except for the tip and the rotation of the upper central incisor, for all the other movements in more than 40% of cases there is a discrepancy between Prescription and Achieved movements error higher than 2°, hence it is frequent in the treatment with aligners, that the Achieved movements were far from the Prescriptions.

Considering that all the assessed movements, except for rotation of the central incisors, showed a performance index greater than 2.0° in absolute value, the error in the other movements may have a visible impact on the clinical outcome. According to our results, torque and rotation corrections showed the same performance index in absolute value for the cumulative analysis, suggesting that B-L inclination and rotation had a comparable level of discrepancy between planned and achieved movements. On the other hand, tip correction showed the worst performance value. Teeth anatomy also affects the accuracy of the movement, indeed while torque and rotations for incisors and canine presented a moderate accuracy, the rotation and the tip of the second molars and second premolars showed a low accuracy. This might be due to two considerations, the first is linked to the shape of the teeth in particular for the second premolars, and second that the second molars usually do not present a more distal teeth, have short crowns and indeed are difficult to move.

Our sample torque of central incisors and canines present 35% of movements with a very high predictability, with an error of less than 1° between Achieved movements and Prescription. This can be due to the protocol used to achieve the torque movements on these teeth, and to the anatomy of the teeth. Besides, rotations have the highest FOPE of movements with less than 2° of error (49.8%) between the Achieved movements and Prescription. Indeed, this movement had the highest predictability in our sample, on all the teeth, a result that is not in agreement with most of the previous investigations on aligners predictability [30,31]. Finally, assessing the iper-performance and the under-performance of all the movements, the study showed a higher frequency of iper-performance in the torque movement, and a higher frequency of under-performance in the tip movement. These results underline the difficulties of aligners to maintain a good root movement. Indeed, the iper-performance in the torque movement might be associated to the uncontrolled tipping during expansion and proclination that increases the buccolingual inclination of the teeth with scarce control of the root; similarly the under-performance in the tip movement might be affected by the difficulties of the aligner to provide a good mesio-distal tipping to the roots [22]. Lombardo et al. (2017) also aimed to evaluate accuracy of F22 aligners [14]. Using our same methodology to analyze the digital models of retrospectively selected patients, they found that the mean predictability of tooth movements achieved was 73.6%. In detail, the highest value of the predictability was met by mesiodistal tipping (82.5%), followed by buccal-lingual tipping (72.9%) and rotations (66.8%). The inhomogeneity of these findings and our results can be due to a different patient’s sample in terms of typology of malocclusions treated and different mechanical performance of the thermoplastic materials used.

To assess the accuracy of CAT, Miller et al. (2003) used palatal rugae as a stable anatomical reference structure to superimpose digital models and then analyze any discrepancy between planned and achieved dental movements using Invisalign© system [32]. However, it should be considered that this methodology is only suitable for the upper arch and the shape and length of rugae can be altered, even if in a slight way, by changes in teeth position such as the result of an orthodontic treatment [33,34]. Another way that can be used to perform models’ superimpositions to evaluate positional changes during CAT is on stable reference teeth that are not moved in the digital set-up. Using this methodology with the software Tooth Measure, Kravitz et al. found that the mean accuracy of tooth movement with Invisalign was 41% [13]. The most accurate movement was lingual constriction (47.1%), and the least accurate movement was extrusion (29.6%). It should be noted, however, that even if some teeth are not moved in the set-up, the anchorage stress can determine a minimal shift on them influencing the reliability of those dental units used as reference structures to perform superimpositions [35]. Finally, also the accuracy of Invisalign system was assessed on 38 patients at the end of treatment, and an average precision of 50% of the predicted movement was achieved [29].

In this study the assessment of tooth movements was performed after stage 15, that is consistent with the time of first refinement (around 8–9 months) reported in treatment with aligners [23]. Considering that for all tooth movements, there is a delay between Prescription and Achieved movements, a check at stage 15 is strongly suggested in order to individuate the delayed movements, and this could also suggest a different strategy in aligners treatment, such as fabricating and sending a first set of only 15–16 aligners at the beginning of the treatment, to avoid the waste of the other aligners that might not fit well.

A major study limitation includes the methodological repeatability in the positioning of selected points, especially the gingival reference points of the FACC, since the gingival margins can change during the orthodontic treatment, and they can be modified in the virtual models by the software that is used to virtually move the teeth. Furthermore, the home compliance in clear aligner therapy could represents a huge risk of bias in our study. On the other hand, the used methodology showed very good intra-operator reproducibility and allowed very accurate measurements of tooth movements in a large sample of patients.

## 5. Conclusions

Summarizing the results of our investigation on the predictability of clear aligners therapy, it can be concluded that:

The tip of the second molars was the movement showing the largest deviation from the right performance and the highest FOPE with an error of more than 4°.

The rotation of the central incisors showed the lowest deviation from the right performance while the torque of central incisors and canines presented the highest FOPE of less than 1°.

Rotation and torque correction showed the same level of performance in absolute value for the overall analysis.

Among the three movements, the tip showed the highest frequency of under-performance, while the torque showed the highest frequency of iper-performance.

Since for some tooth movements a significant difference was found between predicted and achieved positions, an accurate evaluation of the treatment after the 15th aligner is strongly suggested in order to individuate the movements that are not following the digital set-up.

## Figures and Tables

**Figure 1 materials-15-02646-f001:**
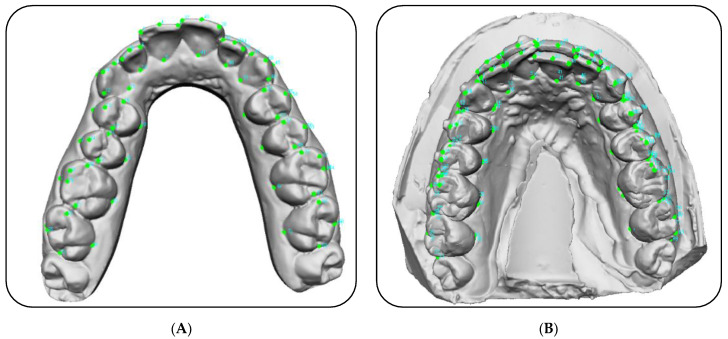
(**A**) Position of the 72 reference points. (**B**) Superimposition with the 72 reference points between T0 and T1 maxillary arch.

**Table 1 materials-15-02646-t001:** Method of error and systematic error for the used analysis.

Tooth	Torque		Tip		Rotation	
	Dahlberg	*t* Test	Dahlberg	*t* Test	Dahlberg	*t* Test
11	0.26	0.864	0.89	0.943	0.75	0.846
12	0.43	0.961	0.78	0.855	3.82	0.117
13	0.51	0.935	0.89	0.946	0.72	0.941
14	0.74	0.997	1.23	0.845	1.35	0.905
15	0.87	0.895	1.14	0.811	1.24	0.979
16	1.03	0.673	1.2	0.433	1.48	0.843
17	0.89	0.740	1.33	0.934	0.83	0.788
21	0.26	0.960	0.69	0.98	0.64	0.909
22	0.27	0.965	0.75	0.763	2.45	0.387
23	0.39	0.891	0.76	0.749	0.99	0.907
24	0.74	0.964	0.61	0.55	1.27	0.806
25	0.89	0.940	1.23	0.92	1.27	0.634
26	0.76	0.961	1.02	0.767	1.11	0.817
27	0.64	0.941	1.47	0.944	1.13	0.835

**Table 2 materials-15-02646-t002:** Descriptive statistics for Prescription Achieved movement.

	Tooth	Number of Teeth	Mean Prescription (°)	Standard Deviation (°)	Mean Achieved Movement (°)	Standard Deviation (°)	*p*-Value
Torque	Central incisors	34	3.2	3.3	3.6	3.6	NS
	Lateral incisors	34	2.8	2.3	2.8	2.8	NS
	Canines	34	1.9	1.5	2.4	2.2	NS
	First premolars	34	2.8	2.6	3.0	2.1	NS
	Second premolars	34	3.0	2.3	3.3	3.1	NS
	First molars	34	2.9	3.1	2.7	2.9	NS
	Second molars	34	2.6	2.1	3.0	2.8	NS
	**All**	**238**	**2.8**	**2.5**	**2.8**	**2.8**	**NS**
Tip	Central incisors	34	2.7	2.4	2.5	2.2	NS
	Lateral incisors	34	2.9	2.7	3.1	2.4	NS
	Canines	34	2.8	2.2	3.3	2.4	NS
	First premolars	34	2.4	1.9	2.8	2.5	NS
	Second premolars	34	3.3	2.9	3.0	2.5	NS
	First molars	34	5.7	11.5	5.7	11.1	NS
	Second molars	34	4.0	2.7	4.1	3.5	NS
	**All**	**238**	**3.4**	**5.0**	**3.5**	**4.9**	**NS**
Rotation	Central incisors	34	4.1	3.9	4.5	4.0	NS
	Lateral incisors	34	3.2	3.4	3.2	3.1	NS
	Canines	34	3.0	2.3	3.5	3.0	NS
	First premolars	34	3.8	3.2	3.5	3.1	NS
	Second premolars	34	4.1	2.7	4.6	3.8	NS
	First molars	34	5.2	10.3	5.3	10.7	NS
	Second molars	34	3.4	2.3	3.0	2.3	NS
	**All**	**238**	**3.8**	**4.8**	**3.9**	**5.0**	**NS**

NS, not significant.

**Table 3 materials-15-02646-t003:** Performance of the achieved movements.

	Tooth	Number of Teeth	Mean Performance (°)	Standard Deviation (°)	FOPE Performance ≤ 1 *n* (%)	FOPE 1 < Performance ≤ 2 *n* (%)	FOPE 2 < Performance ≤ 4 *n* (%)	FOPE Performance > 4 *n* (%)
Torque	Central incisors	34	2.2	1.8	12	6	12	4
					(35.3%)	(17.6%)	(35.3%)	(11.8%)
	Lateral incisors	34	2.3	1.8	7	11	9	7
					(20.6%)	(32.3%)	(26.5%)	(20.6%)
	Canines	34	2.1	1.6	12	6	10	6
					(35.3%)	(17.6%)	(29.4%)	(17.6%)
	First premolars	34	2.1	1.5	10	11	8	5
					(29.4%)	(32.3%)	(23.5%)	(14.7%)
	Second premolars	34	2.7	1.9	10	3	13	8
					(29.4%)	(8.8%)	(38.2%)	(23.5%)
	First molars	34	2.8	2.1	11	2	11	10
					(32.3%)	(5.9%)	(32.3%)	(29.4%)
	Second molars	34	3.2	2.0	6	5	15	8
					(17.6%)	(14.7%)	(44.1%)	(23.5%)
	**All**	**238**	**2.5**	**1.8**	**68**	**44**	**78**	**48**
					**(28.6%)**	**(18.5%)**	**(32.8%)**	**(20.2%)**
Tip	Central incisors	34	2.1	1.8	11	10	8	5
					(32.3%)	(29.4%)	(23.5%)	(14.7%)
	Lateral incisors	34	3.3	2.5	11	3	6	14
					(32.3%)	(8.8%)	(17.6%)	(41.2%)
	Canines	34	2.3	1.9	10	9	9	6
					(29.4%)	(26.5%)	(26.5%)	(17.6%)
	First premolars	34	2.8	1.8	5	11	9	9
					(14.7%)	(32.3%)	(26.5%)	(26.5%)
	Second premolars	34	2.4	1.9	9	9	10	6
					(26.5%)	(26.5%)	(29.4%)	(17.6%)
	First molars	34	3.3	2.4	7	6	10	11
					(20.6%)	(17.6%)	(29.4%)	(32.3%)
	Second molars	34	3.8	2.7	8	4	7	15
					(23.5%)	(11.8%)	(20.6%)	(44.1%)
	**All**	**238**	**2.9**	**2.2**	**61**	**52**	**59**	**66**
					**(25.6%)**	**(21.8%)**	**(24.8%)**	**(27.7%)**
Rotation	Central incisors	34	1.8	1.4	11	14	6	3
					(32.3%)	(41.2%)	(17.6%)	(8.8%)
	Lateral incisors	34	2.3	1.9	11	8	11	4
					(32.3%)	(23.5%)	(32.3%)	(11.8%)
	Canines	34	2.8	1.7	5	7	15	7
					(14.7%)	(20.6%)	(44.1%)	(20.6%)
	First premolars	34	2.6	1.9	8	11	6	9
					(23.5%)	(32.3%)	(17.6%)	(26.5%)
	Second premolars	34	3.2	2.3	5	10	9	10
					(14.7%)	(29.4%)	(26.5%)	(29.4%)
	First molars	34	2.0	1.4	10	5	16	3
					(29.4%)	(14.7%)	(47.0%)	(8.8%)
All	Second molars	34	3.1	1.9	9	4	9	12
					(26.5%)	(11.8%)	(26.5%)	(35.3%)
	**All**	**238**	**2.5**	**1.8**	**59**	**59**	**72**	**48**
					**(24.8%)**	**(24.8%)**	**(30.2%)**	**(20.2%)**
	**Total**	**714**	**2.6**	**2.0**	**188**	**155**	**209**	**162**
					**(26.3%)**	**(21.7%)**	**(29.3%)**	**(22.7%)**

FOPE, Frequencies of Performance Error.

**Table 4 materials-15-02646-t004:** Chi-square test.

	Under-Performance	Right-Performance	Iper-Performance	
TORQUE	60	68	110	238
TIP	112	61	65	238
ROTATION	87	59	92	238
	259	188	267	714

Under-performance (Performance < −1°) Right-performance (−1° ≤ Performance ≤ 1°) Iper-performance (Performance > 1°).

## Data Availability

The data presented in this study are available on request from the corresponding author.

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
