# Peer review of "Evaluation of Tooth Movement Accuracy with Aligners: A Prospective Study"

_materials, 2022, doi:10.3390/ma15072646_

Round 1

Reviewer 1 Report

Dear Author,

Thank you very much for your interesting manuscript.

However, there are some concerns regarding manuscript improvement:

Line 73: Stage 15?

Please explain why did you select stage 15? Does every CAT have the same number of stages? Only in that case, it can be write like that, without explanation.

Is 15 half of the therapy, or near to the end stage of the therapy? I feel that this is also important information.

Line 81: What is specific informed consent?

Also, you did not mention in the section Aim of the study or Materials and Methods that you are going to check the movement of maxillary teeth, exclusively. You only stated it in the first paragraph of Conclusion. Please, consider changing it.

Author Response

Dear Author,

Thank you very much for your interesting manuscript.

However, there are some concerns regarding manuscript improvement:

Line 73: Stage 15?

Please explain why did you select stage 15? Does every CAT have the same number of stages? Only in that case, it can be write like that, without explanation.

Is 15 half of the therapy, or near to the end stage of the therapy? I feel that this is also important information.

We thank the reviewer or this comment. The choice of stage 15 was independent by the number of stages of the treatment. All the included cases were full cases, with more than 24 aligners. Stage 15 was indicated as an average stage where enough movement was performed to be detected, and it is next to the usual time for first refinement request (after 8-9 months of treatment, Hansa et al., 2020).

Hansa I, Semaan SJ, Vaid NR. Clinical outcomes and patient perspectives of Dental Monitoring® GoLive® with Invisalign®-a retrospective cohort study. Prog Orthod. 2020;21:16.

A paragraph was added in the discussion:

In this study the assessment of tooth movements was performed after stage 15, that is consistent with the time of first refinement (8-9 months) reported in treatment with aligners [Hansa et al., 2020]. Considering that for all tooth movements, there is a delay between Prescription and Achieved movements, it is strongly suggested a check at stage 15 in order to individuate the delayed movements, and this could suggest also a different strategy in aligners treatment, such as fabricating and sending a first set of only 15-16 aligners at the beginning of the treatment, to avoid the waste of the other aligners that might not fit well.

Line 81: What is specific informed consent?

A specific informed consent was the informed consent approved by the ethical committee of the University of Naples Federico II, for this study.

Also, you did not mention in the section Aim of the study or Materials and Methods that you are going to check the movement of maxillary teeth, exclusively. You only stated it in the first paragraph of Conclusion. Please, consider changing it.

We thank the reviewer for this comment. We added in the aim of the study that only maxillary teeth were included.

“Hence, the aim of this study was to evaluate the predictability of clear aligners not at the end of treatment but comparing the virtually planned and the achieved tooth movement at the end of the stage 15, exclusively in the maxillary arch.”

This was already present in the materials and methods Line 113,Line 138

Reviewer 2 Report

Thank you for giving me the opportunity to review your paper.

I have some concerns with the use of only one clear aligner system in your study. In your conclusion, you have to remove the name of the clear aligner brand as if you are advertising for them since you have received no funding for your research. I would suggest using at least 2 or 3 different systems and increase your sample size accordingly.

You did not mention the kappa statistics regarding your examiner's performance.

Why did you choose T15 this was not mentioned in your materials and methods?

The conclusion should sum up the results of your study and not to have them in bullets form. 

Author Response

Thank you for giving me the opportunity to review your paper.

I have some concerns with the use of only one clear aligner system in your study. In your conclusion, you have to remove the name of the clear aligner brand as if you are advertising for them since you have received no funding for your research. I would suggest using at least 2 or 3 different systems and increase your sample size accordingly.

We thank the reviewer for this comment. The brand of the clear aligner system was deleted by the conclusions. We also appreciate the reviewer idea and this could be a possible future studies comparing the predictability of different clear aligners system.

You did not mention the kappa statistics regarding your examiner's performance.

The method of error and the systematic error were assessed by means of the Dahlberg’s Formula and a Student’s t test for paired data (p<0.05).

Why did you choose T15 this was not mentioned in your materials and methods?

We thank the reviewer or this comment. Stage 15 was indicated as an average stage where enough movement was performed to be detected, and it is next to the usual time for first refinement request (after 8-9 months of treatment, Hansa et al., 2020).

Hansa I, Semaan SJ, Vaid NR. Clinical outcomes and patient perspectives of Dental Monitoring® GoLive® with Invisalign®-a retrospective cohort study. Prog Orthod. 2020;21:16.

A paragraph was added in the discussion:

In this study the assessment of tooth movements was performed after stage 15, that is consistent with the time of first refinement (8-9 months) reported in treatment with aligners [Hansa et al., 2020]. Considering that for all tooth movements, there is a delay between Prescription and Achieved movements, it is strongly suggested a check at stage 15 in order to individuate the delayed movements, and this could suggest also a different strategy in aligners treatment, such as fabricating and sending a first set of only 15-16 aligners at the beginning of the treatment, to avoid the waste of the other aligners that might not fit well.

The conclusion should sum up the results of your study and not to have them in bullets form.

We thank the reviewer for the comments, the bullet points were eliminated, and conclusion were rephrased.

Summarizing the results of our investigation on the predictability of Clear Aligners therapy, it can be concluded that:

The Tip of the second molars was the movement showing the largest deviation from the right performance and the highest FOPE with an error of more than 4°.

The Rotation of the central incisors showed the lowest deviation from the right performance while the Torque of central incisors and canine presented the highest FOPE of less than 1°.

Rotation and Torque correction showed the same level of Performance in absolute value for the overall analysis.

Among the three movements the tip showed the highest frequency of under-performance, while the torque showed the highest frequency of iper-performance.

Reviewer 3 Report

Dear authors,

the chosen topic is very important for orthodontic practice. 

MAJOR

In introduction it is mentioned:

“...the effectiveness of tooth movement is strictly related not only to the virtual setup but also to the mechanical properties of the thermoplastic materials of aligners and attachment design”.

It should be presented which of the mechanical properties of the thermoplastic materials of aligners are related to the effectiveness of tooth movement.

“Temperature, humidity, salivary enzymes and elastic deformation could influence the physical and chemical properties of clear aligners in the mouth of patients”.

Must be detailed how the temperature, humidity, salivary enzymes and elastic deformation could influence the physical and chemical properties of clear aligners in the mouth of patients.

Ethical approval before the study should be mentioned exactly like this:

 “The study was conducted in accordance with the Declaration of Helsinki, and approved by the Institutional Review Board (or Ethics Committee) of the University of Naples Federico II Prot. 353/19.

The iconography needs to be improved.

Author Response

Dear authors,

the chosen topic is very important for orthodontic practice. 

MAJOR

In introduction it is mentioned:

“...the effectiveness of tooth movement is strictly related not only to the virtual setup but also to the mechanical properties of the thermoplastic materials of aligners and attachment design”.

It should be presented which of the mechanical properties of the thermoplastic materials of aligners are related to the effectiveness of tooth movement.

We thank the reviewer for this comment, a sentence was included in the introduction.

Indeed, hardness, elasticity, resilience and resistance to storage in artificial saliva are related to tooth movements [Tamburrino et al.,, 2020]

“Temperature, humidity, salivary enzymes and elastic deformation could influence the physical and chemical properties of clear aligners in the mouth of patients”.

Must be detailed how the temperature, humidity, salivary enzymes and elastic deformation could influence the physical and chemical properties of clear aligners in the mouth of patients.

We thank the reviewer for this comment, a sentence was included in the introduction.

Furthermore, the use of aligners in the oral cavity exposes the aligner to other factors such as temperature, humidity, salivary enzymes and elastic deformation that could influence the physical and chemical properties of clear aligners in the mouth of patients [19,20]. Aligners are affected by intraoral aging that does not change the chemical composition but deteriorate the mechanical properties. [Gerard Bradley et al., 2016]

Gerard Bradley T, Teske L, Eliades G, Zinelis S, Eliades T. Do the mechanical and chemical properties of InvisalignTM appliances change after use? A retrieval analysis. Eur J Orthod. 2016 Feb;38(1):27-31.

Ethical approval before the study should be mentioned exactly like this:

 “The study was conducted in accordance with the Declaration of Helsinki, and approved by the Institutional Review Board (or Ethics Committee) of the University of Naples Federico II Prot. 353/19.

We thank the reviewer for this comment. The sentence was included in the manuscript.

The iconography needs to be improved.

A new image was added, including the superimposition between t0 and t1 (after 15 steps) maxillary arch, to assess achieved movements.

Reviewer 4 Report

Dear Authors,

To be approved, the description of the number of participants must be clearly described, the more detailed methodology, the relevanceand originality of the study.

Regards

Author Response

Dear Authors,

To be approved, the description of the number of participants must be clearly described, the more detailed methodology, the relevance and originality of the study.

The number of participants were included in the results, changes in the introduction and discussion section were performed.

Introduction

Furthermore, the use of aligners in the oral cavity exposes the aligner to other factors such as temperature, humidity, salivary enzymes and elastic deformation that could influence the physical and chemical properties of clear aligners in the mouth of patients [19,20]. Aligners are affected by intraoral aging that does not change the chemical composition but deteriorate the mechanical properties. [Gerard Bradley et al., 2016]

“Hence, the aim of this study was to evaluate the predictability of clear aligners not at the end of treatment but comparing the virtually planned and the achieved tooth movement at the end of the stage 15, exclusively in the maxillary arch.”

Discussion

In this study the assessment of tooth movements was performed after stage 15, that is consistent with the time of first refinement (8-9 months) reported in treatment with aligners [Hansa et al., 2020]. Considering that for all tooth movements, there is a delay between Prescription and Achieved movements, it is strongly suggested a check at stage 15 in order to individuate the delayed movements, and this could suggest also a different strategy in aligners treatment, such as fabricating and sending a first set of only 15-16 aligners at the beginning of the treatment, to avoid the waste of the other aligners that might not fit well.

Hansa I, Semaan SJ, Vaid NR. Clinical outcomes and patient perspectives of Dental Monitoring® GoLive® with Invisalign®-a retrospective cohort study. Prog Orthod. 2020;21:16.

Conclusions

Summarizing the results of our investigation on the predictability of Clear Aligners therapy, it can be concluded that:

The Tip of the second molars was the movement showing the largest deviation from the right performance and the highest FOPE with an error of more than 4°.

The Rotation of the central incisors showed the lowest deviation from the right performance while the Torque of central incisors and canine presented the highest FOPE of less than 1°.

Rotation and Torque correction showed the same level of Performance in absolute value for the overall analysis.

Among the three movements the tip showed the highest frequency of under-performance, while the torque showed the highest frequency of iper-performance.

Round 2

Reviewer 2 Report

You need to mention why did you choose T15 in your abstract and introduction..

The reader needs to know your purpose for conducting this research and why did you choose the timing of for assessing the success of CAT. 

Author Response

You need to mention why did you choose T15 in your abstract and introduction..

The reader needs to know your purpose for conducting this research and why did you choose the timing of for assessing the success of CAT. 

We thank the reviewer for the comment changes were performed in the abstract and in the introduction

Abstract: This study aimed to evaluate the predictability of CAT, comparing the virtually planned and the achieved tooth movement at the end of the stage 15 that is often the time of first refinement. 

Introduction : A common time for the first refinement is around 8-12 months of treatment,[23] for this reasons in this study was decided to assess the tooth movements after stage 15.

Reviewer 3 Report

Dear authors,

I reviewed the article and I saw that you responded to the requests.

In this new version the article can be published. 

Author Response

We want to thank the reviewer for the comment.